# Improved Synthetic Route of Incorporation of Nanosilicon Species into Phenol-Formaldehyde Resin and Preparation of Novel ZnAl-Layered Double-Hydroxide Hybrid Phenol-Formaldehyde Resin

**DOI:** 10.3390/polym14214684

**Published:** 2022-11-02

**Authors:** Ana Dremelj, Romana Cerc Korošec, Andreja Pondelak, Branka Mušič

**Affiliations:** 1Nanotesla Institut Logatec, Obrtna cona Logatec 4, 1370 Logatec, Slovenia; 2Fakulteta za Kemijo in Kemijsko Tehnologijo, Univerza v Ljubljani, Večna pot 113, 1000 Ljubljana, Slovenia; 3Zavod za Gradbeništvo Slovenije, Dimičeva ulica 12, 1000 Ljubljana, Slovenia

**Keywords:** phenol-formaldehyde hybrid resins, nano-SiO_2_, ZnAl-LDH, composites, glass-transition temperature, mechanical properties

## Abstract

Hybrid phenol-formaldehyde (PF) resins represent one of the most important niche groups of binding systems for composites. New industrial needs, environmental requirements, and price fluctuations have led to further research on materials with enhanced mechanical and thermal properties. The preparation of novel hybrid materials can be achieved by inclusion of various elements or functional groups in the organic polymer phenolic framework. Herein, we report the synthesis and characterization of a PF-based hybrid material with different nanoscale silicone species and ZnAl-layered double hydroxide (LDH). The main goals of this study were to improve the synthetic pathways of hybrid resin, as well as to prepare granulated composite materials and test samples and determine their characterization. Added inorganic species increased the glass-transition temperature by a minimum of 8 °C, which was determined using differential scanning calorimetry (DSC). Rheological properties (melting viscosity and flow distance) of the hybrid resin were measured. The homogeneity of distribution of added species across the organic matrix was evaluated with scanning electron microscopy (SEM). With synthesized new hybrid-binding systems, we prepared different granulated composite materials and evaluated them with the measurements of rheological properties (flow curing characteristics). Tensile strength of samples, prepared from granulated composite material, improved by more than 5%.

## 1. Introduction

Phenol-formaldehyde (PF) resins have had one key role in the manufacturing of polymer composite materials for decades. They are an essential component in the production of paints, adhesives, and coatings, as well as in the automotive and electronic industries [1]. The research field of PF resins is nowadays divided into two streams. The first one is interested in improving specific properties through the incorporation of different organic or inorganic species [2,3,4]. The second one is focused on moving from phenol and formaldehyde to biodegradable or environmentally friendly raw materials [5,6,7].

The main goal of the synthesis of organic–inorganic hybrid materials is the synergy of both materials. While the organic part contributes to toughness, elasticity, and also enables functionalization, the inorganic part provides hardness and thermal and mechanical stability [8]. Silicon compounds and precursors are one of the most represented species for the preparation of various hybrid PF resins [9,10,11,12,13]. It has already been confirmed that the addition of silicon species improves both thermal and mechanical properties [14,15,16]. Most of the hybrid resins were prepared by the sol–gel process, in which the biggest problem represents the phase separation and agglomeration that occur during the synthesis [17,18,19]. 

The preparation of the layered double hydroxide (LDH)-containing hybrids from PF novolac resin is a less-researched area. LDH can be described by a formula [M^2+^_1−x_M^3+^_x_(OH)_2_]^x+^A^n−^_x/n_·mH_2_O, where M^2+^ and M^3+^ represents a divalent or a trivalent metal cation (e.g., Mg^2+^, Zn^2+^, and Al^3+^). A^n−^ is a charge-compensating inorganic or organic anion (e.g., CO_3_^2−^, SO_4_^2−^, NO_3_^−^, and RCO^−^) [20]. Several positive effects have already been noticed when combining LDH with different materials. In biomedicine applications, they can successfully serve as a controlled drug delivery and release system [21]. In the field of polymer materials, it has already been proved that the addition of LDH can improve the mechanical properties and thermal stability of polymers [22].

The aim of this work is to examine the in situ synthesis of nano-SiO_2_ phenol-formaldehyde hybrid resin and the preparation of ZnAl-hybrid phenol-formaldehyde resin. For the nano-SiO_2_ phenol-formaldehyde hybrid resin, we focused on the homogeneity of the system, which is crucial for uniform mechanical properties. We tried to achieve it with different synthetic paths. We also tested the concentration dependence of added species on the selected properties. Individual types of zinc and aluminum species together are very common partners in composite materials, as they facilitate processability and help to improve specific properties [23]. We prepared the hybrid resin of ZnAl-LDH with phenol-formaldehyde resin and characterized it with different analytical methods. We investigated the effects of both added species on glass-transition temperature, melt viscosity, and reactivity of hybrid resins. Based on synthesized hybrid binding systems, we prepared different granulated composite materials and explored their rheological characteristics, the content of volatile organic components, and reactivity. For the evaluation of mechanical properties and dimensional stability, test samples from granulated materials via hot-press machine were prepared.

## 2. Materials and Methods

### 2.1. Materials

Phenol (99+%), formaldehyde (37% aqueous solution stabilized with 7–8% methanol), and oxalic acid (98%) were purchased from Alfa Aesar (Kandel, Germany). 3-aminopropyltriethoxysilane (99%) was purchased from Dynasylan, Evonik (Essen, Germany). The colloidal dispersion of nano-SiO_2_ was prepared according to the process described in the patent SI23390A [24]. ZnAl-layered double hydroxide was prepared as described in Cerc Korošec et al. [25]. For the preparation of the granulated composite material, we used hexamethylenetetramine (HMTA-99%) from Alfa Aesar (Kandel, Germany) as curing agent, milled glass fibers from Lanxess (Cologne, Germany) as reinforcement agent, nontreated and silane-treated kaolin from Imerys (Paris, France) as fillers and lubricant, and release agent and pigment as additives. 

### 2.2. Synthesis and Preparation

#### 2.2.1. Synthesis of Novolac Resin with 3-Aminopropyltriethoxysilane (AMEO)

Phenol, formaldehyde, and oxalic acid were put into the four-necked round-bottom glass flask, equipped with a thermometer, mechanic stirrer, and condenser. The reaction took place at about 100 °C until the concentration of formaldehyde fell below 0.5%, determined according to ISO standard 11402. Then, we started the atmospheric distillation until the temperature of the mixture rose to about 180 °C. At 180 °C, we started with vacuum distillation until the concentration of the free phenol in the reaction mixture fell below 0.5%. The mixture was then cooled to about 130 °C and the 3-aminopropyltriethoxysilane was added. The reaction mixture was then poured into a tray and allowed to cool to room temperature.

#### 2.2.2. Synthesis of Hybrid Novolac Resin with Colloidal Dispersion of Nano-SiO_2_ (COSI)

Phenol, formaldehyde, colloidal dispersion of nano-SiO_2_, and oxalic acid were put into the four-necked round-bottom glass flask, equipped with a thermometer, mechanic stirrer, and condenser. The reaction took place at about 100 °C until the concentration of formaldehyde fell below 0.5%. Then, we started the atmospheric distillation until the temperature of the mixture rose to about 180 °C. At 180 °C, we started with vacuum distillation until the concentration of the free phenol in the reaction mixture fell below 0.5%. The reaction mixture was then poured into a tray and allowed to cool to room temperature.

#### 2.2.3. Preparation of Hybrid Novolac Resin with ZnAl-Layered Double Hydroxide

Phenol, formaldehyde, and oxalic acid were put into the four-necked round-bottom glass flask, equipped with a thermometer, mechanic stirrer, and condenser. The reaction took place at about 100 °C until the concentration of formaldehyde fell below 0.5%. Then, we started the atmospheric distillation until the temperature of the mixture rose to about 180 °C. At 180 °C, we started with vacuum distillation until the concentration of the free phenol in the reaction mixture fell below 0.5%. The reaction mixture was then poured into a tray and allowed to cool to room temperature. ZnAl-layered double hydroxide (LDH) was then added in three different ways. Firstly, the novolac resin was heated to 70 °C and the LDH was added. Secondly and thirdly, the novolac resin and LDH were crushed together and heated to 70 °C.

#### 2.2.4. Preparation of Granulated Composite Materials

The novolac resin or hybrid novolac resin was mixed together with hexamethylenetetramine (HMTA), nontreated and silane-treated kaolin, lubricant, release agent, dye, and milled glass fibers (Figure 1). The mixture was heated to about 75 °C and the water was added. The mixture was then kneaded, cut to small granules, and dried at 86 °C.

### 2.3. Methods of Characterization 

Methods of characterization were used in two different stages of the process. The first stage of characterization took place after the synthesis/preparation of resins and the second stage was completed for the characterization of the granulated composite material.

#### 2.3.1. Characterization of Resins


Karl Fischer titration


Water content was determined with a volumetric titrator (DL38, Mettler Toledo, Greifensee, Switzerland) based on ISO standard 15512:2016 [26]. The reagent for volumetric titration was pyridine free one-component HYDRA POINT composite 5 (J. T. Baker, NJ, USA). The sample mass was 800 ± 20 mg.
Gas chromatography with flame ionization detector

The analysis of free phenol content was performed using gas chromatography with a flame ionization detector (GC-FID, Trace 1300, Thermo Fischer Scientific, Waltham, MA, USA). The method was based on the ISO 8974:2002 [27] standard for determination of residual phenol content by gas chromatography. The separation column used was the Rxi-5Sil MS column (30 m × 0.32 mm × 0.25 µm). For evaluation, Chromeleon^TM^ Chromatography Data System software was used.
Flow-distance test

Flow-distance test of phenolic resin powder is specified with ISO standard 8619:2003 [28], according to which some internal adjustments were made: the height of the pellet was 5 ± 0.2 mm and the diameter was 14.5 ± 0.3 mm. Flow distance is dependent on the reactivity and on the melt viscosity of the resin.
Melt-viscosity analysis

Melt viscosities (80 ± 2 mg of sample) were measured using a cone and plate rotational viscometer (RM100 CP2000, LAMY Rheology, Champagne au Mont d’Or, France) at shear rate 600 s^−1^. The temperature was 150 °C and time of measurement was 60 s.
Fourier-transform infrared spectroscopy with attenuated total reflectance (FTIR-ATR)

FTIR measurements were performed on Bruker Alpha II FTIR-ATR Instrument (Bruker, Billerica, MA, USA). For evaluation, OPUS 7.8 software was used. The scanning range was between 400–4000 cm^−1^.
Differential scanning calorimetry (DSC)

Glass-transition temperature was determined using a DSC823^e^ Mettler Toledo Instrument (Mettler Toledo, Greifensee, Switzerland). We used 40 µL aluminum pans with pierced lids. Samples (10–15 mg) were analyzed at the temperature range from 25 to 160 °C at a heating rate 10 K/min under air atmosphere. The data collection and analysis were performed using the program package STARe Software 16.2 (STAR^e^, Software 16.2, Mettler-Toledo GmbH, (2019)).
Scanning electron microscopy (SEM)

Samples were investigated using a scanning electron microscope (JEOL JSM-IT500, Japan) with SEM operational software. The resin samples of approximately 10 × 5 × 3 mm^3^ were prepared using epoxy resin (EpoFix Resin, Struers, Denmark) and hardener (EpoFix Hardener, Struers, Denmark) and were polished. Samples were placed on double-sided carbon tape and, additionally, were Au/Pd (n(Au):n(Pd) = 80:20)-coated using (Quroum Q150T ES). The thickness of the Au/Pd coating was 20 nm. The images were carried out in high-vacuum mode; a working distance of 10 mm, an accelerated voltage of 15 kV, and a secondary electron detector (SED) were used.

#### 2.3.2. Characterization of Granulated Composite Material


Volatile organic compounds


The test samples were heated and held at 168 °C using a halogen moisture analyzer apparatus (HG 63, Moisture Analyzer, Mettler Toledo, Greifensee, Switzerland). Aluminum sample pans were filled with 13 ± 0.1 g and the analysis time was 12 min.
Rheology analysis

The flow-curing behaviors of PMCs were determined using a torque rheometer (Brabender Plastograph EC plus, MB30 mixer, Brabender GmbH & Co. KG, Duisburg, Germany). The 25 g of test samples were loaded into the mixer bowl at 140 °C. The speed of the triangle-shaped mixing blades was 30 rpm. The measurements shown on the plastogram represent the time dependence of the torque at a given temperature with respect to the changes in material structure. The evaluation was made with Brabender data correlation software according to standard DIN 53 764 [29]. 

#### 2.3.3. Characterization of Hot-Pressed Cylindrical Samples


Density


The density of cured cylindrical samples was measured in water at room temperature. We used a precision analytical balance (XP205, Delta Range, Mettler Toledo, Greifensee, Switzerland) with the associated density kit.
Mechanical test

The mechanical characterization of cured final cylindrical test samples was carried out using a static tensile machine (Z100, Zwick/Roell, Ulm, Germany) at a speed of 86 mm/min. The tensile strength was measured on hollow cylindrical samples. The height of the sample was 15.00 ± 0.05 mm (H), outer diameter 20.00 ± 0.05 mm (D), and inner diameter 12.00 ± 0.05 mm (d).

## 3. Results

The samples we prepared followed each other in the following order. Sample R1 represents the basic resin without additives. Samples R2–R4 were prepared with the addition of AMEO, COSI was added to samples R5–R7, and samples R8–R10 contained ZnAl-LDH. In all of them, the lowest concentration of additive was 1.5 wt% and the highest was 3 wt%. From all hybrid resins (R1–R10), we prepared granulated composite material (GC1–GC10), in which 25 wt% represents resin with hardener, 30 wt% fillers with additives, and 25 wt% fibres. 

### 3.1. Water Content and Free Phenol Content

The water content and free phenol content of none of the measured samples exceeded 0.5%, which was set for two of the basic goals in all syntheses (Table 1). 

### 3.2. Flow Distance and Melt Viscosity 

Flow distance and melt viscosity of the resins varied with the content of the water, free phenol, and concentration of additions of different species in the polymer matrix. Since the concentrations of water and free phenol content in all synthesized hybrid systems were in the same range, they did not play a decisive role in the changing melt viscosities and flow lengths. The hardener concentration was also the same in all samples. It is observed that melt viscosity and flow length are reciprocally correlated. The results show that the melting viscosity of the hybrid resins increases with the addition of inorganic species, while the flow distance decreases. Both are also dependent on the concentration of added species (Table 1). 

### 3.3. Fourier-Transform Infrared Spectroscopy with Attenuated Total Reflectance (FTIR-ATR)

There were no significant differences between all of the synthesized resins. The concentration of added species was small and the phenol-formaldehyde matrix signals were very pronounced. We can see representative peaks for novolac resins (wavenumber (cm^−1^)): 504.9; 753.4; 814.6; 1097.0; 1169.0; 1204.9; 1331.9; 1433.2; 1506.4; 1594.1; 3287.2 [30]. A peak indicating a presence of nano-SiO_2_ occurred at 1715 ± 20 cm^−1^, while the peak representative for ZnAl-LDH occurred at 1354 ± 3 cm^−1^
Figure 2). 

### 3.4. Differential Scanning Calorimetry (DSC) 

The DSC measurements were performed in order to determine the glass-transition temperatures (Tg) of the synthesized resins. The glass transition was determined using the STAR^e^ software function. The sp-line was determined as the baseline. The glass-transition temperature is the temperature at which increased molecular mobility results in changing the glassy state of the resin into a rubbery or melted one. The results show that with all additives, the glass-transition temperatures increased compared to the basic resin (Figure 3). A trend can be observed in how, at a concentration of 1.5 and 2% of any chosen additive (R2–R6 and R8–R10), the glass-transition temperature moves in the same range by approximately 8 °C higher than the basic resin. A larger deviation occurs with the addition of 3% COSI (R7), when the temperature of the glass transition rises by an additional 6 °C (Table 1).

### 3.5. Scanning Electron Microscopy (SEM)

Using a scanning electron microscope, we determined how homogeneously nano-SiO_2_ and ZnAl-LDH were distributed along the organic matrix during one-pot synthesis. Nanoparticles are known to have a high tendency to aggregate and agglomerate, which was also confirmed in this case. A bright area, nearly in the middle of Figure 4b,c represents aggregates of colloidal SiO_2_ and ZnAl-LDH particles, around which PF resin is organized. Nevertheless, it was further shown that partial agglomeration does not affect the deterioration of mechanical properties of the final hot-pressed cylindrical samples prepared from granulated composite material.

### 3.6. Volatile Organic Compounds

The content of the volatile organic components in the granulated composite material, which was estimated on the basis of preliminary tests not to affect the dimensional changes of the test samples after the curing process, was set between 2.7 and 3.3 wt%. The value of the volatile organic components of all composites prepared from hybrid resins ranged between 2.8 and 3.3 wt%. 

### 3.7. Rheology Analysis

The flow-curing behavior plastograms illustrate the torque measurement over the period of the test duration and are a reflection of structural changes in the granulated material. The characteristic values of the measurements are the torque minimum (B) and the residence (tV) and reaction (tR) times. The torque minimum determines the melt viscosity of a granulated sample before the curing reaction. The residence time determines the time duration of material in a molten state before starting the curing reaction. After reaching the maximum torque peak and the torque stabilizes, the greater part of the curing reaction is completed, which defines the reaction time. The results show a slight increase in minimum viscosity for all granulated composite materials prepared from resins with additives, but no significant deviation occurs for any. The residence and reaction time of composite material prepared from hybrid resins with aminosilane are slightly shorter, which means that the curing reaction takes place a little faster than with other composites. The addition of inorganic additives in resin has no noteworthy effect on the crosslinking rate (Table 2, Figure 5).

### 3.8. Density

The density of a hot-pressed cylindrical samples was measured before and after the postcuring process. In all test samples the precuring density was around 1900 ± 20 g/cm^3^, indicating that the hybrid resin does not affect the density of the final cylindrical samples. However, a trend is observed, in which the density of the postcuring samples was about 1.0% lower than the density before postcuring. 

### 3.9. Mechanical Test 

The dimensional measurements of the cylindrical samples before and after the postcuring processes showed very small dimensional changes (about 0.3%) in all three measured parameters. It was observed that the outer diameter and height slightly decreased, while the inner diameter slightly increased. With the selected composition of the composite material, the addition of aminosilane had no additional effect on the tensile strength, which can be partly attributed to the fact that one of the fillers was already coated with aminosilane [31]. The tensile strengths of all composites, prepared with hybrid resins, which contained COSI, were 7–14% greater than in basic resin composites. The addition of ZnAl-LDH to the resin increased the tensile strength of the composite by just under 6% (Table 2). 

## 4. Discussion

The addition of nano-SiO_2_ and ZnAl-LDH compared to aminosilane and basic resin without additives, the influence of selected additives on the properties of hybrid resins, and the granulated composite material prepared from them were experimentally investigated.

The water content has a significant role in plasticity of novolac phenolic resins and even more so affects the melt viscosity. Its content, from 0.1 to 3%, can reduce melt viscosity by up to 90%. When the water content increases, the flow distance decreases due to the higher reactivity of the resin with the hardener, despite the fact that the melt viscosity drops. Free phenol content plays a similar role to water content, except that its effect is less rigorous [1,32]. 

In the FTIR spectrum of the hybrid resins with nano-SiO_2_, we expected a peak at approximately 1100 and 810 cm^−1^ [33]. However, the concentration of the additive was too low, and at the same time, the peaks of the basic resin were prominent in this region, so that they overlapped. The peak, representative of ZnAl-LDH (1357 cm^−1^) is characteristic of a free carbonate anion and represents the carbonate stretching mode [34,35].

The glass-transition temperature varies with the addition of inorganic additives and their concentration. Inorganic additives, dispersed in a matrix, can suppress the local mobility of the polymer material and, therefore, increase its glass-transition temperature.

The large specific surface area of nanoparticles and the attractive forces between them lead to the formation of agglomerates. This is greatly helped by the high viscosity of the polymer matrix, which prevents their optimal dispersion. Nevertheless, the agglomerate regions were not so pronounced as to affect the mechanical properties of the test samples [31].

## 5. Conclusions

The key findings of the research on the one-pot synthesis of nano-SiO_2_ hybrid resins and solid–solid mechanochemical milling addition of ZnAl-LDH are as follows: The formation of the hybrid resins via in situ synthesis with nano-SiO_2_ was successful, but the aggregation of nanoparticles into larger agglomerates was partially confirmed. The formation of the agglomerate did not affect the final properties of the resin, which were nevertheless homogenous, as well as the properties of the prepared composite material from this hybrid resin.

Melt viscosities, flow distances, and glass-transition temperatures change markedly with the addition of an inorganic additive. With the addition of nano-SiO_2_, individual properties depend on the concentration of the additive itself, while with aminosilane and ZnAl-LDH the concentration does not play a significant role. Crosslinking rate of granulated composite material does not depend on the addition of nano-SiO_2_ and ZnAl-LDH to the hybrid resin, nor on their concentration, while the addition of the aminosilane slightly accelerates the reaction.

The mechanical properties of hot-pressed cylindrical samples were remarkably improved in the composite, prepared with the nano-SiO_2_ (7–14%) and slightly improved with the addition of ZnAl-LDH (6%). The mechanical properties of hybrid resins with aminosilane were in the same range as for the basic resin composites.

An overall comparison shows that the hybrid resin with the addition of nano-SiO_2_ is the most suitable for use, since its synthesis is the simplest, most economically efficient, and least energy consuming. The granulated composite material from this resin also exhibits the best mechanical properties and does not affect the viscosity or crosslinking reaction. However, it should be emphasized that the further optimization of the in situ synthesis is necessary, and the formation of agglomerates should be reduced or even eliminated.

## Figures and Tables

**Figure 1 polymers-14-04684-f001:**
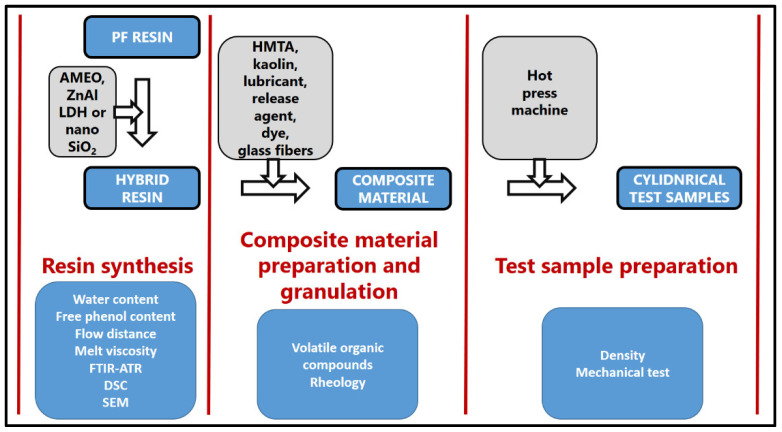
Preparation stages: from resin synthesis to hot-pressed samples.

**Figure 2 polymers-14-04684-f002:**
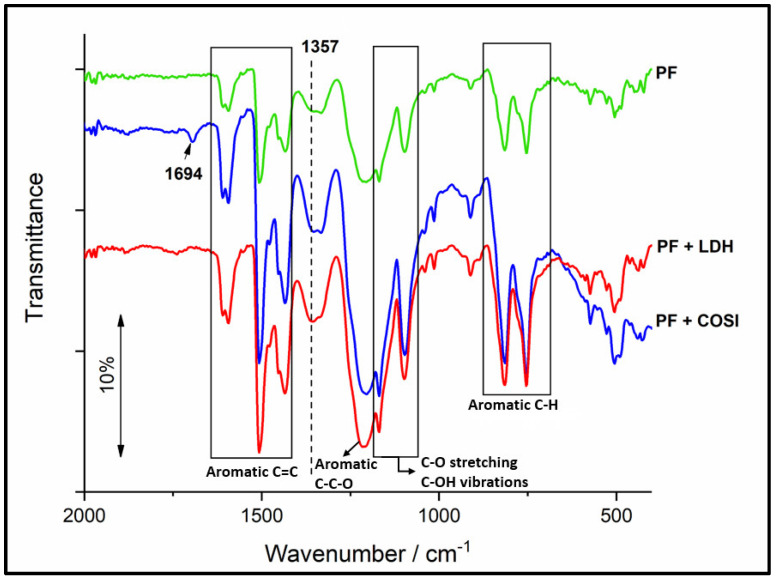
FTIR spectra of the unmodified resin (green curve), with addition of colloidal silica (blue curve) and modified with ZnAl-LDH (red curve).

**Figure 3 polymers-14-04684-f003:**
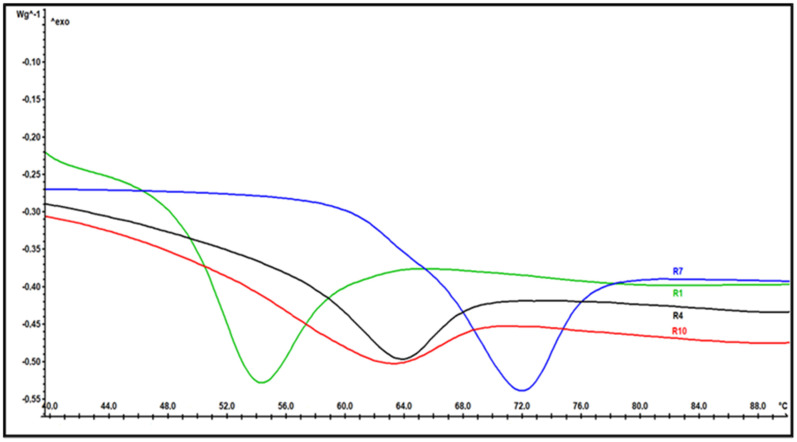
Differential scanning calorimetry (DSC) thermograms of four synthesized resins (basic–R1, with aminosilane–R4, with COSI–R7, and with ZnAl-LDH–R10).

**Figure 4 polymers-14-04684-f004:**
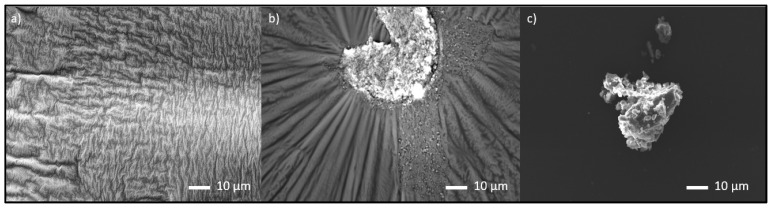
Scanning electron microscope images: (**a**) basic resin with no added species; (**b**) resin with nano-SiO_2_; (**c**) resin with ZnAl-LDH.

**Figure 5 polymers-14-04684-f005:**
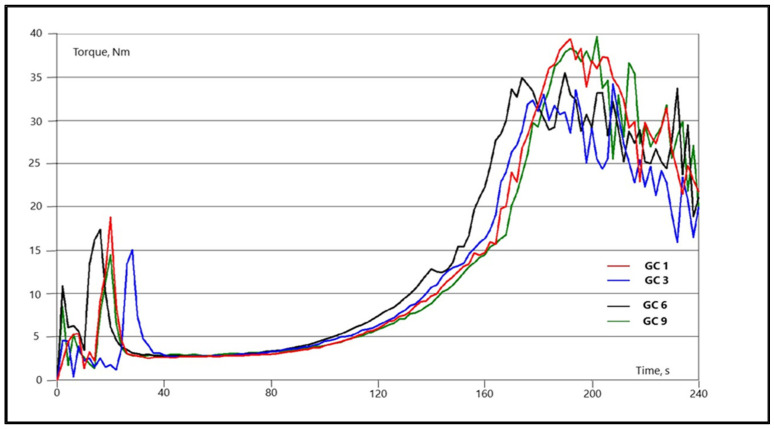
Examples of flow-curing behaviors of some prepared GC composites.

**Table 1 polymers-14-04684-t001:** Water and free phenol content, melt viscosities, flow distances, and glass-transition temperatures of all synthesized resins.

Resin Type	Resin	Concentration of Added Species (%)	Content of H_2_O (%)	Free Phenol Content (%)	Melt Viscosity (mPas)	Flow Distance (mm)	Tg (°C)
**Basic**	R1	/	0.19	0.25	2821	80	49.8
**Hybrid with AMEO**	R2	1.5	0.09	0.45	2360	94	57.9
	R3	2.0	0.19	0.20	2154	97	58.6
	R4	3.0	0.17	0.48	2034	99	59.0
**Hybrid with COSI**	R5	1.5	0.17	0.23	3997	49	58.8
	R6	2.0	0.17	0.49	5461	33	59.4
	R7	3.0	0.19	0.16	6232	24	65.8
**Hybrid with ZnAl-LDH**	R8	1.5	0.19	0.25	4047	47	58.7
	R9	2.0	0.19	0.25	4205	47	59.9
	R10	3.0	0.19	0.25	4480	48	59.7

**Table 2 polymers-14-04684-t002:** Flow-curing behavior properties and mechanical properties of GC material and hot-pressed cylindrical test samples.

Resin Type	Resin	Composite Material	Torque Minimum; B [Nm]	Residence Time; tV [s]	Reaction Time; tR [s]	Tensile Strength [N]
**Basic**	R1	GC1	2.5	92	160	1678
**Hybrid with AMEO**	R2	GC2	2.7	87	147	1670
	R3	GC3	2.8	82	141	1686
	R4	GC4	2.9	71	133	1685
**Hybrid with COSI**	R5	GC5	2.7	94	165	1806
	R6	GC6	2.7	91	150	1946
	R7	GC7	2.7	97	163	1925
**Hybrid with ZnAl** **LDH**	R8	GC8	2.6	102	167	1775
	R9	GC9	2.7	103	163	1769
	R10	GC10	2.6	99	160	1787

## Data Availability

The data presented in this study are available on request from the corresponding author.

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
