# Peer review of "Improved Synthetic Route of Incorporation of Nanosilicon Species into Phenol-Formaldehyde Resin and Preparation of Novel ZnAl-Layered Double-Hydroxide Hybrid Phenol-Formaldehyde Resin"

_polymers, 2022, doi:10.3390/polym14214684_

Round 1

Reviewer 1 Report

In this manuscript, the authors propose the improvement of synthetic route of incorporation of nano silicon species into phenol-formaldehyde resin. The authors sustain that was prepared hybrid resin of ZnAl LDH with phenol-formaldehyde resin and then were evaluated the effects of added inorganic species on the glass transition temperature using differential scanning calorimetry (DSC) and rheological properties (melting viscosity and flow distance) of hybrid resin. The homogeneity of distribution of added species across the organic matrix was evaluated with scanning electron microscopy (SEM).

In my opinion the manuscript proposed for publication should be rewritten with a clear highlighting of the sections: Materials and Methods, Results, Discussion, Conclusions (for example the transition from section 2 (Results) to section 3 (Discussion) is not highlighted). My recommendation is to also improve the interpretation of the experimental results and highlight the degree of novelty of this study.

 General comments

·       The abstract and keywords are meaningful.

·       The manuscript contains descriptions of studies that have been planned/carried out but the interpretation of experimental results must be improved.

o    There is no information regarding the description of the samples, which additive was chosen, in what proportion: which is the difference between R1, R2 .... R10, also GC1, GC2 … GC10? Compositionally? Weight? It should be explained!

o   the characterization of the synthesized materials is not presented. For example, the synthesis of LDH is mentioned, but nothing about the characterization of the synthesized powder.

·       The effects of added inorganic species on the glass transition temperature using differential scanning calorimetry (DSC) and rheological properties (melting viscosity and flow distance) of hybrid resin must better explained.

·       The manuscript is quite well related to existing literature but can be improved. There are more standards indicated without specifying the number, the authority...

·       There are not formulated conclusion for this study. I know that this section is not mandatory but can be added to the manuscript if the discussion is unusually long or complex.

The specific comments are as follows:

Line 50-53: it is necessary to correct the writing of formulas and ionic species by using superscript and subscript în all paper: [M2+1-50 xM3+x(OH)2]x+An–x/n.mH2O, where M2+ and M3+ (e.g. Mg2+, Zn2+, Al3+), (e.g. CO32–, SO42–, NO3, RCO–)

Line 80 „in the patent SI23390A” – the bibliographic source should be indicated

Line 148: „according to the standard” - which standard ??

Line 206: „Figure 2. Drawing of hot-pressed cylindrical sample used for mechanical test” - it is not significant for the description of the mechanical test - I propose its elimination

Author Response

Dear reviewer.

Best regards, Ana and Branka

Reviewer 2 Report

Revision for Polymers (ISSN 2073-4360)

Manuscript ID: polymers-1973356

Title: Improved synthetic route of incorporation of nano silicon species into phenol-formaldehyde resin and preparation of novel ZnAl layered double hydroxide hybrid phenol-formaldehyde resin

List of authors: Ana Dremelj * , Romana Cerc Korošec , Andreja Pondelak , Branka Mušič

This research paper deals with the preparation of hybrid phenol-formaldehyde (PF) resins and their characterization. The authors investigated the influence of the incorporation of nanoscale silicone species and ZnAl layered double hydroxide on the thermal and rheological properties of PF based hybrid materials. Also, the authors evaluated the effect of such incorporation on the morphology and the distribution of the added species throughout the organic matrix. I consider the research work a valuable activity and it can open many future prospective in the field of hybrid phenol-formaldehyde resins and their application. As the subject is interesting, I am willing to recommend minor revisions with pending manuscript decision. I will gladly be able to review the modified manuscript once the following points have been fully addressed:

1) Page 1, lines 13-28 - I would suggest to insert some quantitative data in the abstract to highlight the main achievements.

2) Page 2, line 46 - I would suggest to specify which types of sol-gel approaches are used to prepare hybrid organic-inorganic composites.

3) Page 2, line 48 - I would suggest to add some additional references to support this statement:

- Gilman, J. W., Kashiwagi, T., Nyden, M., Brown, J. E. T., Jackson, C. L., Lomakin, S., ... & Manias, E. (1999). Flammability studies of polymer layered silicate nanocomposites: polyolefin, epoxy, and vinyl ester resins. Chemistry and technology of polymer additives, 14, 249-265.

- Branda, F., Bifulco, A., Jehnichen, D., Parida, D., Pauer, R., Passaro, J., ... & Durante, M. (2021). Structure and bottom-up formation mechanism of multisheet silica-based nanoparticles formed in an epoxy matrix through an in situ process. Langmuir, 37(29), 8886-8893.

- Judeinstein, P., & Sanchez, C. (1996). Hybrid organic–inorganic materials: a land of multidisciplinarity. Journal of Materials Chemistry, 6(4), 511-525.

- Zhi, M., Liu, Q., Chen, H., Chen, X., Feng, S., & He, Y. (2019). Thermal stability and flame retardancy properties of epoxy resin modified with functionalized graphene oxide containing phosphorus and silicon elements. ACS omega, 4(6), 10975-10984.

4) Page 2, lines 52-53 - The chemical formulas should be revised with correct superscripts and subscripts.

5) Page 2, line 80 - I think that the patent should be cited as a reference in the paper.

6) Page 3, line 96 - Would be possible to add a small scheme of the preparation route to better clarify each step for the reader?

7) Page 3, line 124 - I would suggest to add more details to the scheme.

8) Page 4, line 131 - The authors should add details regarding the software used for the different characterization and the type of version for the programs.

9) Page 6, lines 226-228 - Is it not also reported a wavenumber of 1100 cm-1 for nano silica? In the case of hybrid silica (not completely polymerized), a broad peak at 1070 cm-1 is also reported in the literature. The authors should have a look at the following paper:

- Mascia, L., Prezzi, L., & Lavorgna, M. (2005). Peculiarities in the solvent absorption characteristics of epoxy‐siloxane hybrids. Polymer Engineering & Science, 45(8), 1039-1048.

10) Page 6, line 230 (Figure 3) - I would suggest to uniform the base line for the spectra and add some indicators for the specific wavenumber related to a particular functional group.

11) Page 6, lines 234-237 - Could the authors mention which method/procedure has been used to determine the glass transition temperature?

12) Page 6, lines 239-241 - Could the authors provide an explanation about this increase of the glass transition temperature?

13) Page 7, line 243 - I would suggest to uniform the style of all the figures throughout the paper.

14) Page 7, lines 251-254 - The aggregation does not occur in all the cases. The authors should provide an explanation about the occurrence of such aggregation phenomena. Also, the authors should explain the reason why such aggregation does not affect the mechanical properties.

15) Page 8, lines 263-267 - Please provide if these % are expressed in weight or molar composition.

16) Page 9, lines 296-299 - I would suggest to add some references supporting these statements. Also, the comparison with similar systems in the literature could better support this evidence.

17) Page 10, line 307 - I would suggest to reduce the “Conclusions” at main highlights and achievements.

Author Response

(The authors gave the same response as above.)

Round 2

Author Response

Thank you very much for your prompt reply.